# A Six-Year Retrospective Study of Microbiological Characteristics and Antimicrobial Resistance in Specimens from a Tertiary Hospital’s Surgical Ward

**DOI:** 10.3390/antibiotics12030490

**Published:** 2023-03-01

**Authors:** Petros Ioannou, Sofia Maraki, Dimitra Koumaki, Georgios A. Manios, Vasiliki Koumaki, Dimitrios Kassotakis, Georgios V. Zacharopoulos, Diamantis P. Kofteridis, Andreas Manios, Eelco de Bree

**Affiliations:** 1School of Medicine, University of Crete, 71003 Heraklion, Greece; 2Internal Medicine Department, University Hospital of Heraklion, 71110 Heraklion, Greece; 3Department of Clinical Microbiology, University Hospital of Heraklion, 71110 Heraklion, Greece; 4Department of Dermatology, University Hospital of Heraklion, 71110 Heraklion, Greece; 5Department of Computer Science and Biomedical Informatics, University of Thessaly, 38221 Lamia, Greece; 6Department of Microbiology, School of Medicine, National and Kapodistrian University of Athens, 15772 Athens, Greece; 7Department of Surgical Oncology, University Hospital of Heraklion, 71110 Heraklion, Greece

**Keywords:** surgical infection, microbiology, antimicrobial resistance, *Acinetobacter*, Enterobacterales, *Klebsiella*, *Escherichia*, *Pseudomonas*

## Abstract

Surgery has revolutionized the practice of medicine by allowing the treatment of conditions amenable to conservative medical management with some of them pathophysiologically involving the prevalence of pathogenic microorganisms. On the other hand, infections such as surgical site infections or urinary tract infections may complicate patients hospitalized in surgical wards leading to considerable morbidity, mortality, and increased healthcare-associated costs. The aim of this study was to present the microbiological characteristics and antimicrobial resistance of all isolates identified in microbiological specimens from a surgical ward of a tertiary hospital in Greece during a six-year period. Only specimens that yielded at least one microorganism were included in the analysis. In total, 1459 strains in 789 positive cultures were isolated. The most common sample sent to the microbiology department was pus from surgical wounds. The most common pathogens among all 1459 strains isolated were Enterobacterales at 33% (*n* = 482), however, the most common genus was *Enterococcus* at 22.3% (*n* = 326). Antimicrobial resistance against third-generation cephalosporins was 23% (*n* = 111/482) among Enterobacterales, while, the rate of vancomycin-resistant enterococci (VRE) was 18.5% (*n* = 60/324) among *Enterococcus* species and was increasing in the last years of the study. Antimicrobial resistance of *Acinetobacter baumannii* to carbapenems was 68.8% (*n* = 11/16), which was lower than the corresponding rate in other wards in Greece. The antimicrobial resistance rates noted herein raise questions regarding the appropriateness of currently suggested antimicrobials in guidelines and imply that a revision could be required. Practicing clinicians should always be aware of local microbiological data that allow the selection of appropriate antimicrobials for the management of infections. Finally, the increasing rates of VRE noted herein mandate further actions from the point of infection control and antimicrobial stewardship.

## 1. Introduction

Even though surgery has revolutionized the practice of medicine by allowing treatment of conditions amenable to conservative medical management such as appendicitis or cholecystitis, and has introduced surgical therapies leading to cures for conditions previously considered untreatable, such as gastric or colon cancer, infections may complicate the course of patients undergoing surgical procedures leading to considerable morbidity, mortality, and increased healthcare-associated costs [1,2,3,4,5,6,7]. Moreover, several pathological conditions involving infectious processes may occur in patients that may require surgical management [8,9,10,11]. It is widely accepted that awareness of microbiological characteristics and antimicrobial resistance at a local, national, and global level can lead to improved decision-making regarding the choice of antimicrobials to be used for the treatment of infections by avoiding unnecessary use of broader spectrum antimicrobials than needed, without depriving patients of appropriate antimicrobial coverage [12,13]. Thus, adequate knowledge of the microbiological characteristics and antimicrobial resistance of pathogens commonly encountered in patients hospitalized in surgical wards is of utmost importance for surgeons, since it allows for a better selection of antimicrobial compounds to be used in everyday surgical practice.

Surgical site infections (SSIs), for example, are commonly complicating surgery and are associated with notable morbidity [14]. Their microbiology more commonly includes *Staphylococcus aureus*, *Enterococcus* spp., coagulase-negative staphylococci, and *Escherichia coli* [15]. Antimicrobials with presumed activity against these pathogens in the cases of such infections are known; however, increasing rates of antimicrobial resistance are noted for SSIs and other surgical infections leading to increased rates of antimicrobial treatment failure, and increased hospitalization duration, hospital costs, morbidity, and mortality [16,17]. Moreover, multi-drug-resistant (MDR) pathogens that are relatively frequent causes of hospital-acquired infections may also be the cause of surgical infections, such as SSIs. For example, in a recent study in Italy, SSIs by MDR pathogens were associated with a higher cost and increased rates of postoperative complications [16]. Furthermore, the problem of antimicrobial resistance also raises questions about what appropriate surgical antimicrobial prophylaxis should be [18,19,20].

Although there are published data on antimicrobial resistance in Greece, there is a paucity of data about antimicrobial resistance in surgical wards specifically, as the majority of the data are at the hospital level or may refer to wards in general, making no discrimination between medical and surgical wards, even though differences could occur between these two different settings [21,22].

The aim of the present study was to present the microbiological characteristics and antimicrobial resistance of all isolates identified in microbiological specimens obtained from a surgical ward of a tertiary hospital in Greece during a six-year period, that includes the emergence of the COVID-19 pandemic, and identify any differences or trends that may have developed at that time period.

## 2. Materials and Methods

### 2.1. Study Type and Ethics Approval

This is a retrospective single-center study including data regarding all isolates from all types of cultures sent to the Microbiology Department from patients hospitalized from 2016 to 2021 in the Department of Surgical Oncology of the University Hospital of Heraklion, Heraklion, Greece, a tertiary hospital with a capacity of 771 beds. All data were retrieved retrospectively from the database of the Department of Microbiology and were then evaluated. Data from cultures were included if the culture was positive for the growth of at least one microorganism. The only exclusion criterion was not yielding any microorganisms in the cultures. Data collected and evaluated included the type of sample that yielded a positive culture, the date the culture was collected, the microorganism identified, and the antimicrobial resistance of the isolated microorganisms.

The conduction of the study was approved by the Institutional Review Board of the University Hospital of Heraklion.

### 2.2. Sample Collection, Transport, and Processing

Blood, lower respiratory tract specimens, urine, pus and exudates, and other biological specimens were collected from patients admitted to the Department of Surgical Oncology of the University Hospital of Heraklion. Blood was sent in blood culture bottles. Lower respiratory tract specimens and urine were collected in appropriate sterile containers. Pus and exudates were collected by using a sterile cotton swab, which was immediately placed in Amies transport medium (bioMérieux SA, Marcy L’Étoile, France). The samples were promptly transported to the Microbiology Laboratory for further processing.

For each specimen wet mount preparations, Gram-stained smears and cultures were carried out following the laboratory protocols. For the isolation of bacterial pathogens, specimens were inoculated onto a variety of ready-to-use general enriched, selective, and differential culture media (all products of bioMérieux SA) and incubated in different incubation conditions. Identification of bacterial species was performed by standard biochemical assays and the Vitek 2 automated system and confirmed using matrix-assisted laser desorption time of flight mass spectrometry (MALDI-TOF MS) (version 3.2) (both products of bioMérieux SA). The Vitek 2 automated system was also used for the antibiotic susceptibility testing and the results were interpreted according to the 2021 Clinical and Laboratory Standards Institute (CLSI) criteria [23]. As quality control strains, *Escherichia coli* ATCC 25922, *Pseudomonas aeruginosa* ATCC 27853, *S. aureus* ATCC 25923, and *E. faecalis* ATCC 29212 were used.

### 2.3. Statistics

Descriptive statistics were performed with GraphPad Prism 6.0 (GraphPad Software, Inc., San Diego, CA, USA). Qualitative data were presented as counts and percentages. Statistical analysis of qualitative data was performed through contingency analysis with a chi-squared test. All tests were two-tailed, and *p*-values < 0.05 were considered to be significant.

## 3. Results

### 3.1. Types of Cultures and Microbiological Characteristics

In total, 1459 microorganisms were isolated from the 789 positive cultures that had been sent to the Microbiology Department of the hospital during the six-year period of the study. The type of cultures along with the number of isolated microorganisms is shown in Table 1, while Appendix A shows the number of isolates identified per year during the study period, as well as detailed microbiological characteristics and the distribution of species among the different years of the study. Among all pathogens, Gram-positive bacteria were the most commonly isolated, since they consisted of 675 strains (46.3%) of all isolated microorganisms, with *Enterococcus* spp. being the most common among them, followed by coagulase-negative *Staphylococcus* and *Streptococcus* spp. Gram-negative bacteria were slightly less frequent than Gram-positive bacteria, consisting of 655 (44.9%) strains with Enterobacterales being the most common among them, with *Escherichia coli* being the predominant one. It is of note that in the present study, Enterobacterales are presented together, even though they consist of different genera because they share common phenotypic characteristics in terms of antimicrobial resistance. Fungi were isolated less frequently, with *Candida* spp. being the only genus identified. Table 2 shows the distribution of the most common pathogens isolated from the patients’ specimens.

### 3.2. Trends of Microbiological Characteristics during the Period of the Study

Enterobacterales were the most common pathogens isolated from specimens of patients hospitalized in the surgical ward, however, at the gender level, *Enterococcus* spp. were the most common isolated pathogen and they showed a trend for increased identification in 2019 and 2020 that was not continued in 2021. The second most commonly isolated pathogen was *Escherichia* spp. (which was *E. coli* in all cases) and their trend showed a decrease that coincided with the increase in *Enterococcus* spp. in 2019 and 2020 but did not continue in 2021. The third most commonly isolated pathogen was *Candida* spp. and their isolation rate among all isolated microorganisms remained almost stable across the study period. *Klebsiella* spp. had a slightly increasing trend during the study period, while *Pseudomonas* spp., *Staphylococcus aureus*, and *Acinetobacter* spp. (which was *A. baumannii* in 16 out of 17 cases) remained relatively stable during the study period. Figure 1 shows the number of isolated microorganisms in regard to the year during the study period and their rate among all isolated microorganisms. Moreover, Table 2 shows the number of microorganisms isolated in the pre-COVID-19 and the post-COVID-19 era. For the majority of microorganisms, there were no statistically significant differences. However, in the post-COVID-19 era, there were more *Klebsiella* and *Corynebacterium* strains as well as fewer *Proteus* strains isolated compared to the pre-COVID-19 era.

### 3.3. Antimicrobial Resistance

Antimicrobial resistance data were collected for all isolated species. Among Enterobacterales, resistance to ampicillin, ceftriaxone, ciprofloxacin, trimethoprim and sulfamethoxazole, piperacillin and tazobactam, colistin, tigecycline, gentamicin, and meropenem was 71.8%, 23.0%, 21.2%, 20.1%, 18.0%, 14.8%, 14.5%, 8.2%, and 6.7%, respectively. More specifically, regarding *Escherichia coli*, resistance to ampicillin, ciprofloxacin, trimethoprim and sulfamethoxazole, ceftriaxone, gentamicin, piperacillin and tazobactam, meropenem, tigecycline, and colistin was 52.0%, 29.8%, 28.3%, 15.2%, 10.5%, 8.1%, 0.9%, 0.5%, and 0%, respectively. Regarding *Klebsiella* spp., resistance to ampicillin, piperacillin and tazobactam, ceftriaxone, ciprofloxacin, meropenem, trimethoprim, and sulfamethoxazole, tigecycline, colistin, and gentamicin was 100%, 37.7%, 34.6%, 33.3%, 30.8%, 21.7%, 11.9%, 10.4%, and 10.3%, respectively. For *Pseudomonas* spp., resistance to ciprofloxacin, ceftazidime, meropenem, piperacillin and tazobactam, colistin, and gentamicin was 22.0%, 14.6%, 10.1%, 8.2%, 1.1%, and 1.1%, respectively. As for *A. baumannii*, resistance to ceftriaxone, meropenem, piperacillin and tazobactam, ciprofloxacin, trimethoprim and sulfamethoxazole, gentamicin, colistin, and tigecycline was 68.8%, 68.8%, 66.7%, 61.5%, 61.5%, 53.3%, 18.8%, and 15.4%, respectively.

For Gram-positive bacteria, *S. aureus* resistance to penicillin, levofloxacin, oxacillin, clindamycin, trimethoprim and sulfamethoxazole, tigecycline, and vancomycin was 81.4%, 37.0%, 34.9%, 25.6%, 2.6%, 0%, and 0%, respectively. Among *Enterococcus* spp., *E. faecalis* comprised exactly 50.0% of strains and its resistance to clindamycin, moxifloxacin, ampicillin, vancomycin, and tigecycline was 100%, 31.8%, 4.9%, 3.7%, and 0%, respectively. *E. faecium* comprised 39.6% of strains and its resistance to clindamycin, moxifloxacin, ampicillin, vancomycin, and tigecycline was 100%, 81.8%, 79.5%, 33.9%, and 3.3%, respectively. Antimicrobial resistance to ampicillin and vancomycin increased in the later years of the study period.

For *Candida* spp., which were the only fungal species identified, antifungal resistance to fluconazole, caspofungin, micafungin, voriconazole, and amphotericin B was 8.0%, 1.6%, 1.6%, 0.8%, and 0%, respectively. Figure 2 shows the antimicrobial resistance of major clinically significant microorganisms in regard to the year during the study period. Among the most commonly isolated microorganisms, and more specifically, *Enterococcus* spp., *S. aureus*, Enterobacterales, *Pseudomonas* spp., *A. baumannii*, and *Candida* spp., there was almost no difference in terms of antimicrobial resistance for the most clinically relevant antimicrobials during the period of the study with one notable exception: an increase in the detection of vancomycin-resistant *Enterococcus* (VRE) in the last years of the study. Appendix A shows the antimicrobial resistance patterns of the most commonly identified species in the present study.

## 4. Discussion

This study presents data regarding all microorganisms isolated in a surgical ward of a tertiary hospital in Greece during a six-year period. More specifically, Enterobacterales were the most common pathogens, while *Enterococcus* was the most commonly identified gender. There were some differences in terms of microbiological characteristics after the onset of the COVID-19 pandemic with the most notable being an increase in the number of microorganisms isolated as well as an increase in isolation of *Klebsiella* and *Corynebacterium* strains and a reduction in isolation of *Proteus* strains. Data regarding antimicrobial resistance were recorded for all microorganisms and are presented for the major types of antimicrobials in total and in regard to the year during the study period, with almost no differences in the most commonly identified species.

The most common type of sample sent for cultures was pus from the surgical site, implying that SSIs may be among the most common infectious processes in a surgical ward, even though the present study did not aim to quantify or identify the most common infections in a surgical ward. SSIs are very common surgical infections and are among the most frequently encountered hospital-acquired infections (HAIs). More specifically, SSIs may occur in up to 20% of patients after surgery, even though this rate varies greatly in regard to factors such as the type of surgery, surgeon’s experience, data collection, and the criteria used for definitions [15,24]. A recent systematic review studied the impact of SSIs on healthcare costs and outcomes of patients in European countries and found that there is a significant financial burden in European countries due to SSIs that also affect the quality of life of affected patients and increase their morbidity and mortality [24]. The second most common type of sample yielding positive cultures was pus and peritoneal fluid, while, other types of specimens, such as urine or blood were less commonly found among the positive cultures. This may reflect the lower likelihood of positive blood cultures being positive among patients admitted in surgical wards, and also the lower probability of patients having a urinary tract infection (UTI) while hospitalized in a surgical ward. For example, among patients admitted for cholecystitis, blood culture positivity was 31.1% among patients from whom a blood culture had been drawn [25]. For patients with acute appendicitis, this rate may be even lower. For example, in a recent study, only 11.1% of patients with acute appendicitis from whom a blood culture had been drawn eventually had a positive blood culture [26]. Importantly, in both studies, a blood culture was obtained from only a proportion of all patients [25,26].

The most common microorganisms identified in specimens in the present study were *E. coli*, *E. faecalis*, *E. faecium*, *P. aeruginosa*, *Candida* spp., and Enterobacterales other than *E. coli* such as *Enterobacter* and *Klebsiella* spp., while other Gram-positive bacteria such as coagulase-negative staphylococci and *S. aureus* were also frequently identified. In an older study regarding the epidemiology and microbiological characteristics of SSIs, *S. aureus*, *P. aeruginosa*, *E. coli*, *S. epidermidis*, and *E. faecalis* were the most commonly isolated microorganisms. Microbiological characteristics in the present study may differ because the samples herein also included other types of positive cultures, such as urine and blood. The data shown in the present study are closer to those shown in a more recent study involving critically ill patients with SSIs where the most common pathogens identified were *E. coli*, *P. aeruginosa*, *C. albicans*, coagulase-negative staphylococci, and *E. faecalis* [27].

Antimicrobial resistance is a global health problem that is associated with significant morbidity and financial burden [28]. Exposure to antimicrobials has been traditionally considered a classic factor associated with an increased likelihood of developing colonization or infection by drug-resistant microorganisms [29]. In a wider context, antimicrobial use at a patient level is not the sole factor associated with antimicrobial resistance, since the majority of antimicrobials used globally have to do with animal use, mostly for livestock [30]. Thus, the efforts to reduce antimicrobial resistance through the reduction of antimicrobial use should also take into account this use of antimicrobials. However, the spread of resistant strains and resistance genes is also an important contributing factor, so it has been postulated that the reduction of antimicrobial consumption through the implementation of antimicrobial stewardship programs may not be sufficient to control antimicrobial resistance [28].

Regarding antimicrobial resistance in the present study, Enterobacterales showed significant resistance to ceftriaxone, a third-generation cephalosporin, while *Enterococcus* spp. have intrinsic resistance to cephalosporins [31,32,33]. This is of great importance, since, based on the results of the present study, intra-abdominal infections caused by Enterobacterales or *Enterococcus* spp. may not show adequate clinical responses if treatment includes such a cephalosporin. The latest guidelines by the Infectious Diseases Society of America (IDSA) as well as other authors in more recent reviews suggest using a cephalosporin (along with metronidazole) in mild to medium severity complicated intra-abdominal infections; however, local microbiological data should always be considered in every case where an infection is suspected and an antimicrobial agent is to be selected [34,35]. Thus, the data regarding microbiological characteristics and antimicrobial resistance presented by the present study raise an issue of guideline applicability and appropriateness in the local setting. Moreover, VRE, a pathogen of increasing clinical importance due to limited therapeutic options, was very frequent in the present study. Even though the current data cannot discriminate between colonization and true infection, a positive culture with such a pathogen does imply that the patient is colonized and could develop an infection with this pathogen. However, there is controversy regarding whether the isolation of this pathogen in samples of patients with intra-abdominal infections is of true clinical importance [36]. Interestingly, in a recent multicenter study in Korea and another study in Spain regarding intra-abdominal infections, Gram-negative bacteria were also the most commonly isolated pathogens, with *E. coli* being the most common, while in the Korean study *Enterococcus* spp. were the most common Gram-positive bacteria [37,38]. Importantly, high resistance of *E. coli* and *Klebsiella* spp. to ceftriaxone (higher than 30% for *E. coli*) was noted, as in the case of the present study, even though the pathogens presented herein were not only from samples from patients with intra-abdominal infections [37]. In another study from China, *E. coli* and *Klebsiella* spp. also presented decreased antimicrobial susceptibility to ceftriaxone, as more than 50% of *E. coli* strains and about 30% of *Klebsiella* spp. were extended-spectrum beta-lactamase (ESBL)-producing strains, thus, having no susceptibility to ceftriaxone [39]. The abovementioned data imply that the current clinical practice guidelines may be based on older data on antimicrobial resistance and should be updated based on more current data. Interestingly, in the same Korean study, VRE strains consisted of more than 20% *Enterococcus* strains, as was similar to the present study [37].

Quinolone resistance among Enterobacterales was higher in the Korean study compared to the data presented in the present study, as in the Korean study, about 45% of *E. coli* strains and about 14% of *Klebsiella* spp. were resistant to ciprofloxacin. In the study from China, resistance to ciprofloxacin was about 40% in non-ESBL-producing *E. coli*, and about 80% in ESBL-producing *E. coli*, while, for *Klebsiella* spp. these rates were 15% in non-ESBL-producing strains and 60% in ESBL-producing strains [39]. According to the data presented herein, up to 30% of Enterobacterales were resistant to ciprofloxacin, which is also worrisome given that the abovementioned IDSA guidelines for mild-to-moderate intra-abdominal infections suggest quinolones as first-line agents for intra-abdominal infections which are known to be predominantly caused by Enterobacterales [34]. In the present study, increasing identification of *Klebsiella* spp. was noted after the onset of the COVID-19 pandemic, but without an obvious increase in antimicrobial resistance.

*Pseudomonas* resistance to the piperacillin and tazobactam combination was high in the study with intra-abdominal infections in Korea, as about 40% of strains were resistant [37]. In a study with *Pseudomonas* strains identified from patients with intra-abdominal infections in Spain, resistance to piperacillin and tazobactam was about 20% in community-acquired and 43% in hospital-acquired intra-abdominal infections [38]. In the present study, such antimicrobial resistance was lower than 20%. Resistance to ciprofloxacin was 15% in the Korean study, 30% for community-acquired, and 36% in hospital-acquired intra-abdominal infections in the Spanish study, while in the present study this rate was 22% [37,38].

Regarding *A. baumannii*, antimicrobial resistance was high; however, when comparing with data from other wards of the same hospital and other hospitals in Greece, where resistance to carbapenems is 100%, the *A. baumannii* strains isolated from specimens in this surgical ward seemed to be more susceptible to antimicrobials [22,40]. In a point-prevalence study performed in the same hospital in Heraklion, as well as in other seven public hospitals in Crete, in a geographically isolated area within Greece, antimicrobial resistance of *A. baumannii* to carbapenems was 100%; however, in the current study, only 69% of the *A. baumannii* strains were resistant to meropenem [22]. This may imply that effective infection control practices in a ward may protect against the in-hospital spread of antimicrobial resistance even if predominance of a pan-drug-resistant strain in other wards of the same hospital has occurred.

Rates of resistance of *S. aureus* to antimicrobials were similar to those in other infections and wards. More specifically, methicillin-resistant *S. aureus* was found in 35% of strains in the present study, which is comparable to the rate found in other settings, infections, and colonization states in Greece which is between 25% and 50% [41,42,43].

The data of this study provide important information that could be of use to clinicians, either surgeons or infectious diseases practitioners. Since the choice of antimicrobials for a suspected infection should be based on local microbiological data, this type of study will always remain of critical importance to allow guidance for the choice of appropriate antimicrobial treatment for patients in such need. This can be seen in two ways. First of all, knowledge of increased antimicrobial resistance of the most common microbial strains causing infections in everyday clinical practice in surgery can allow treatment with broad-spectrum antimicrobials that provide adequate coverage against the offending pathogens. On the other hand, information regarding antimicrobial resistance dictates the need for the implementation of an antimicrobial stewardship program that would allow more rational use of antimicrobials to reduce their unnecessary use, thus, minimizing the risk of the development of antimicrobial resistance. Furthermore, these data encourage the implementation of infection control practices that may have limited the spread of resistant pathogens, such as *A. baumannii* in this surgical ward, and also sets higher targets, such as limiting the spread of VRE strains to other patients, and lowering the high rates noted in herein.

One important aspect of the present study has to do with the fact that it involves data before and after the onset of the COVID-19 pandemic. There are several studies addressing the effect of the pandemic on surgery and surgical care. For example, in a study from the Netherlands, a reduction was noted in the number of surgeries performed that was greatest for patients without cancer. However, when surgery was performed, it appeared to be performed safely, with similar rates of complications and mortality, a shorter hospital stay, and fewer admissions to the ICU [44]. Interestingly, even though one would anticipate that due to fewer surgical interventions during the pandemic, leading to a lower number of infectious complications and, thus, to a lower number of microorganisms isolated as well, this was not observed during this study. Contrary to that, there was a slight increase of microbial isolates found during 2020 and 2021 compared to the previous years of the study, even though this does not necessarily imply a higher hospital-acquired infection rate, since this was not possible to evaluate based on the study design. This could have several explanations. First of all, even though programmed surgeries may have been postponed due to the pressure of the pandemic on the healthcare system, acute conditions such as cholecystitis, appendicitis, and liver abscesses would continue to occur at least at the same rate and subsequently lead to similar admissions and emergency surgeries. Moreover, oncological operations were performed nearly at the same level as before the pandemic. Furthermore, hospital-acquired infections would continue to occur in surgical patients despite increased infection prevention and control measures as was recently shown in a study in Australia [45]. More specifically, the risk of an individual developing a surgical site infection may not have been affected during the pandemic [46].

Interestingly, there was almost no difference in terms of antimicrobial resistance during the present study among *S. aureus*, Enterobacterales, *Pseudomonas* spp., *A. baumannii*, and *Candida* spp. which were the most commonly isolated microorganisms. However, there was a notable increase in the detection of VRE in the last years of the study. This may be of particular concern for surgeons, since even though *Enterococcus* spp. have been commonly considered a relatively innocent bystander, this pattern of antimicrobial resistance to vancomycin and ampicillin (which is almost universal for *E. faecium*—the most common strain among VRE strains) leaves few therapeutic options for these patients [36,47,48,49]. These data on the increasing prevalence of VRE strains are in line with other recent evidence from Greece showing a high carriage of VRE strains either in terms of colonization or in terms of infections in the hospital setting in recent years [22,50]. Furthermore, the increase in the detection of VRE mandates the activation of infection control and antimicrobial stewardship in a local and possibly, a more generalized setting, since infection control measures could allow the reduction of the spread of these pathogens to other patients [51]. On the other hand, implementation of appropriate antimicrobial stewardship practices could allow for a reduction of unnecessary vancomycin use that could lead to a reduction in future detection of VRE strains, even though in some studies, application of antimicrobial stewardship programs had minor effects in the isolation of VRE strains [52,53,54].

This study has some notable limitations. First of all, it is a single-center study, and this implies that the results should be read cautiously, as they are anticipated to represent the microbiological data and antimicrobial resistance patterns of a specific geographic region. Secondly, there are no data regarding which of the isolated microorganisms truly represent infection, and which refer only to colonization, as these are microbiological data only. Furthermore, for the same reason, there are no data regarding patients’ treatment or mortality. Finally, some of the microorganisms were very few, as in the case of *Acinetobacter*, thus, the data regarding antimicrobial resistance may not be reliable enough to draw safe conclusions.

## 5. Conclusions

The present single-center study presented the microbiological characteristics and antimicrobial resistance of all microbial isolates that were identified in all types of microbiological specimens sent from a surgical ward of a tertiary hospital in Greece during a six-year period. The most common type of microbiological sample that was positive was pus from surgical wounds. The most common pathogens were Enterobacterales, however, the most common gender was *Enterococcus*. Antimicrobial resistance was high for third-generation cephalosporins among Enterobacterales, while the rate of VRE was also very high among *Enterococcus* spp. and showed an increasing trend. The antimicrobial resistance rates noted herein raise questions regarding the appropriateness of currently suggested antimicrobials in guidelines and imply that a revision could be required. Practicing clinicians should always be aware of local microbiological data that allow the selection of appropriate antimicrobials for the management of infections. Furthermore, the increase in isolation of VRE strains warrants further action with active surveillance by infection control services and investigation of the possibility of implementation of antimicrobial stewardship measures for the reduction of unnecessary vancomycin use that could lead to a reduction of resistance to vancomycin.

## Figures and Tables

**Figure 1 antibiotics-12-00490-f001:**
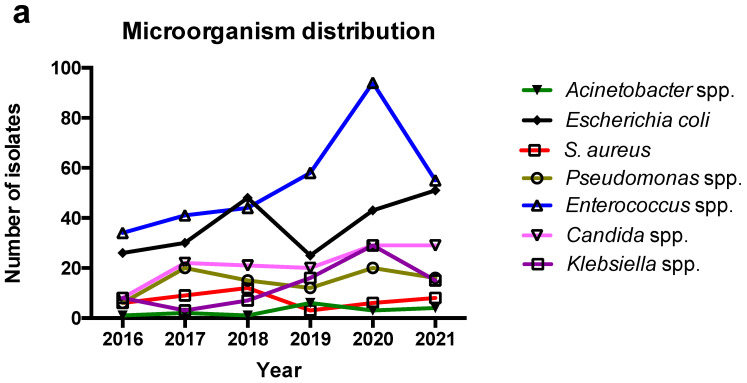
Number and rate of isolated microorganisms from patients’ samples in regard to the year. The number of microorganisms is shown in (**a**), while, the percent of microorganisms among all isolated pathogens is shown in (**b**).

**Figure 2 antibiotics-12-00490-f002:**
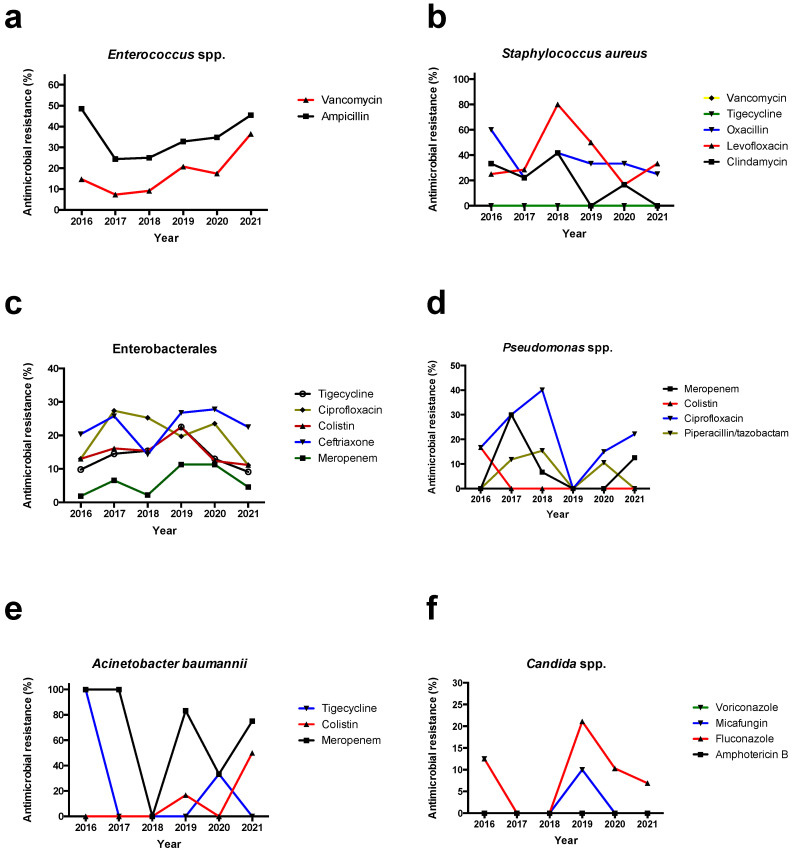
Antimicrobial resistance of isolated microorganisms in regard to the year during the study period. Data shown represent antimicrobial resistance of (**a**) *Enterococcus* spp., (**b**) *Staphylococcus aureus*, (**c**) Enterobacterales, (**d**) *Pseudomonas* spp., (**e**) *Acinetobacter* spp., and (**f**) *Candida* spp. Regarding *Acinetobacter* spp. (all isolates were *A. baumannii*), only 16 strains were isolated.

**Table 1 antibiotics-12-00490-t001:** Types of cultures sent to microbiology and number of isolates identified.

Type of Sample	Number of Isolates (%)
Pus from surgical trauma	384 (26.3)
Pus from abscess	205 (14.1)
Pus (not specified)	203 (13.9)
Peritoneal fluid	174 (11.9)
Urine	104 (7.1)
Blood	96 (6.6)
Fluid (not specified)	82 (5.6)
Bile	81 (5.6)
Tissue	57 (3.9)
Vascular catheter	36 (2.5)
Unknown	8 (0.5)
Bronchial secretions	5 (0.3)
Graft	4 (0.3)
Prosthetic materials	4 (0.3)
Catheter (not specified)	3 (0.2)
Drain catheter	3 (0.2)
Vaginal	2 (0.1)
Stool	2 (0.1)
Bone	2 (0.1)
Sputum	2 (0.1)
Duodenal fluid	1 (0.1)
Pleural fluid	1 (0.1)
All samples	1459 (100)

**Table 2 antibiotics-12-00490-t002:** Microbiological characteristics of the most common pathogens isolated from patients’ samples.

Pathogen	2016–2021 (%)	Pre-COVID-19 (%)	Post-COVID-19 (%)	*p*-Value
**Gram-positives**	**675 (46.26)**	**391 (46.27)**	**284 (45.88)**	**0.9155**
*Enterococcus* spp.	326 (22.34)	177 (21.02)	149 (23.90)	0.1622
Coagulase-negative *Staphylococcus*	152 (10.42)	94 (11.12)	58 (6.86)	0.2985
*Streptococcus* spp.	105 (7.20)	65 (7.69)	40 (4.73)	0.4124
*Staphylococcus aureus*	44 (2.95)	30 (3.57)	14 (2.32)	0.1660
*Corynebacterium* spp.	10 (0.69)	1 (0.12)	9 (1.07)	0.0026
*Clostridium* spp.	9 (0.62)	6 (0.71)	3 (0.36)	0.3168
**Gram-negatives**	**655 (44.89)**	**383 (45.33)**	**277 (44.75)**	**0.8318**
Enterobacterales ***	482 (32.95)	278 (32.79)	204 (33.10)	0.9103
*Escherichia* coli	223 (15.38)	129 (15.22)	94 (15.53)	1
*Enterobacter* spp.	90 (6.17)	54 (6.39)	36 (4.26)	0.7413
*Pseudomonas* spp.	89 (6.04)	53 (6.23)	36 (5.85)	0.8248
*Klebsiella* spp.	78 (5.53)	34 (4.03)	44 (7.02)	0.0094
*Proteus* spp.	40 (2.74)	32 (3.79)	8 (0.95)	0.0033
*Citrobacter* spp.	28 (1.92)	19 (2.25)	9 (1.07)	0.3359
*Morganella* morganii	16 (1.10)	8 (0.95)	8 (1.29)	0.6138
*Acinetobacter* spp.	17 (1.16)	10 (1.15)	7 (1.16)	1
**Fungi ****	**129 (8.84)**	**71 (8.40)**	**58 (9.37)**	**0.5155**
*Candida albicans*	72 (4.93)	36 (4.26)	36 (5.82)	0.1802
*Candida glabrata*	20 (1.37)	10 (1.18)	10 (1.62)	0.5017
*Candida tropicalis*	18 (1.23)	14 (1.66)	4 (0.65)	0.0960
*Candida parapsilosis*	11 (0.75)	6 (0.71)	5 (0.81)	1
All microorganisms	1459 (100)	845 (100)	619 (100)	

* The term Enterobacterales is used here instead of individual genera and species to represent a group of different Gram-negative bacteria that share common phenotypic characteristics in terms of antimicrobial resistance. ** All fungi were *Candida* spp. More details can be seen in Appendix A.

## Data Availability

The data presented in this study are available on request from the corresponding authors.

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
