# Peer review of "A Six-Year Retrospective Study of Microbiological Characteristics and Antimicrobial Resistance in Specimens from a Tertiary Hospital’s Surgical Ward"

_antibiotics, 2023, doi:10.3390/antibiotics12030490_

Round 1

Reviewer 1 Report

Due to absence the of demographic data for patients and its statistical correlation between microbes and resistance to antibiotics. So, I think this manuscript does not contain any novelty and academic originality is weak, and thus this may not meet the standard of the journal

Thank you

Author Response

Due to absence the of demographic data for patients and its statistical correlation between microbes and resistance to antibiotics. So, I think this manuscript does not contain any novelty and academic originality is weak, and thus this may not meet the standard of the journal

Thank you

Response: Thanks for the time spent reviewing this manuscript. However, we feel that articles providing information on microbiology and antimicrobial susceptibility are also important, as they could provide important information that could affect guidelines about empirical antimicrobial treatment in surgical infections.

Reviewer 2 Report

The authors presented a manuscript entitled "Microbiology and antimicrobial resistance in a tertiary hospital's surgical ward- a six-year retrospective study". They evaluated the microbiology and the resistance phenotypes of 1459 isolates form 789 cultures obtained from several surgeries. The "n" evaluated is very good an statistically significant. The topic is relevant to the field and the article is well written and presented, I have only a few observations which are listed below:

- Please check the entire document for the word "Enterobacterales", it corresponds to the order and should not be written in italics.

- Lines 44-45, the bacterial genus and species names must be italicized.

- In Table 1, Please shift the word "Fungi" into bold shape to be homogenized with the other microorganisms groups. Please check the number of isolates, it does not corresponds with totals. Enterobacterales are considered as an individual specie? several species below belong to Enterobacterales order. The same occurs with Fungi and Candida spp. I suggest to modify the presentation of the groups you refer to in the table, is not so clear at first view.

-Line 101. You mentioned that all the Escherichia spp. are in fact E. coli, the same occurs with Acinetobacter spp. I understand why you analyze the results as "spp." but it seems not necessary. I suggest to use E. coli or A. baumanii (in fact you use S. aureus)

- L106. Please add a comma "," after Pseudomonas spp.

- Figure 1. Please use E. coli or A. baumanii instead "spp." as I mentioned before. Please introduce an extra space between "S.aureus" in panel A).

L162. Please erase the extra word "in"

Author Response

The authors presented a manuscript entitled "Microbiology and antimicrobial resistance in a tertiary hospital's surgical ward- a six-year retrospective study". They evaluated the microbiology and the resistance phenotypes of 1459 isolates form 789 cultures obtained from several surgeries. The "n" evaluated is very good an statistically significant. The topic is relevant to the field and the article is well written and presented, I have only a few observations which are listed below:

- Please check the entire document for the word "Enterobacterales", it corresponds to the order and should not be written in italics.

Response: Thanks for the comment. We changed that throughout the text.

- Lines 44-45, the bacterial genus and species names must be italicized.

Response: Done

- In Table 1, Please shift the word "Fungi" into bold shape to be homogenized with the other microorganisms groups. Please check the number of isolates, it does not corresponds with totals. Enterobacterales are considered as an individual specie? several species below belong to Enterobacterales order. The same occurs with Fungi and Candida spp. I suggest to modify the presentation of the groups you refer to in the table, is not so clear at first view.

Response: Indeed, the numbers do not match, since there are several species that occurred relatively rarely, thus, they were not put in Table 2. Furthermore, we used the term Enterobacterales, that indeed refers to many bacterial genera from a practical standpoint, so to include all the gram-negative pathogens that may share common phenotypes in terms of antimicrobial resistance. It is true that a full table with all pathogens encountered should be provided. Thus, we made such a full table and included that in the supplementary table for the reader to be able to see all the different microorganisms that were encountered during this study. This can be found as Table S1. If the reviewer and the editor prefer to add Table S1 in the main text, this could be done in a future revision, if requested.

-Line 101. You mentioned that all the Escherichia spp. are in fact E. coli, the same occurs with Acinetobacter spp. I understand why you analyze the results as "spp." but it seems not necessary. I suggest to use E. coli or A. baumanii (in fact you use S. aureus)

Response: We changed that. As for Acinetobacter, we found one strain of A. lwoffi, and we added that in the analysis. So, we kept Acinetobacter spp. This can be seen in the revised version of the manuscript.

- L106. Please add a comma "," after Pseudomonas spp.

Response: Done

- Figure 1. Please use E. coli or A. baumanii instead "spp." as I mentioned before. Please introduce an extra space between "S.aureus" in panel A).

Response: We did the changes suggested with the exception of Acinetobacter, as we found one strain of Acinetobacter lwoffi, so that Acinetobacter spp. now makes sense.

L162. Please erase the extra word "in"

Response: Done

Reviewer 3 Report

Review Report.

Dear Editor,

Thank you very much for giving me an opportunity to review the article titled

(1.)  Title: Microbiology and antimicrobial resistance in a tertiary hospital’s surgical ward – a six-year retrospective study

OK

(2.)  Introduction: needs modification please make introduction clear :

Suggestion:  mention  surgical site infection (SSI) pathogens with references

Highlight acquisition of antibiotic resistance in (SSI) and write impact of unnoticed SSI resistance with references.

(3.)  Materials & Methods:   I recommend authors to add more information and made clear methods. It needs major revision.

Study design and setting , clinical criteria for selection and data analysis and statistical tests

(4.)  Results  many times to was used please rearrange whole paragraph. Line (114-144)

(5.)    Discussion: its poorly written and needs a major modifications, please include the following

1.      Discuss & compare main factors that contribute drug resistance

2.      Explain Staphylococcus aureus and Pseudomonas aeruginosa and  E. coli and klebsiella in distribution of SSI and Resistance rate.

(6 ).       References:

Ok

Author Response

Dear Editor,

 Thank you very much for giving me an opportunity to review the article titled

(1.)  Title: Microbiology and antimicrobial resistance in a tertiary hospital’s surgical ward – a six-year retrospective study

OK

Response: Thanks for the comment.

(2.)  Introduction: needs modification please make introduction clear :

Suggestion:  mention  surgical site infection (SSI) pathogens with references

Highlight acquisition of antibiotic resistance in (SSI) and write impact of unnoticed SSI resistance with references.

Response: Thanks for the comment. We modified the introduction section by adding a paragraph with the requested information to make the introduction easier for the reader to read. This can be seen in the introduction section of the revised manuscript.

(3.)  Materials & Methods:   I recommend authors to add more information and made clear methods. It needs major revision.

Study design and setting , clinical criteria for selection and data analysis and statistical tests

Response: Thanks for the comment. We performed some modifications of the methods section in order to make it clearer to the reader. The statistics were not sophisticated and were performed with GraphPad Prism. The type of the study was retrospective single-center. There were no specific criteria for inclusion. All positive cultures were included in the analysis. We have modified the previous methods section and added a small paragraph at the end to allow the reader to understand this information. This can be seen in the revised version of the manuscript.

(4.)  Results  many times to was used please rearrange whole paragraph. Line (114-144)

Response: The reviewer has a point in that in subsection 3.3, the word ‘to’ is mentioned many times. We have radically changed that subsection and made those paragraphs easier for the reader to understand, as can be seen in the revised version of the manuscript. Furthermore, upon request from another reviewer, we made some tables in the supplementary material to allow the reader to more easily understand the ‘antibiogram’ of these pathogens. More specifically, we have added data on antimicrobial resistance on all the pathogens that appeared at least 8 times during the study period, thus, representing at least 0.5% of all samples, and this can be seen in the 40 pages of Table S2 that shows this information.

(5.)    Discussion: its poorly written and needs a major modifications, please include the following

  1. Discuss & compare main factors that contribute drug resistance

Response: Thanks for the comment. We have modified the discussion section and added a paragraph about the origins of antimicrobial resistance. This can be seen in the discussion section of the revised manuscript. However, based on the comments of most of the reviewers and the editor, we chose not to radically change the discussion section, but we added text in order to implement the modifications requested by the reviewer.

  1. Explain Staphylococcus aureus and Pseudomonas aeruginosa and E. coli and klebsiella in distribution of SSI and Resistance rate.

Response: Thanks. We have added a paragraph in the discussion section where we mention the most frequent pathogens in our study and compare it to the microorganism distribution noted in other studies that describe patients with SSIs.

(6 ).       References:

Ok

Response: Thanks.

Reviewer 4 Report

The submitted manuscript presents “Microbiology and antimicrobial resistance in a tertiary hospital’s surgical ward – a six-year retrospective study”. The author explained the epidemiology of AMR before and after COVID-19, which is good for researchers and physicians to treat infectious diseases and find out alternative therapeutic measures.  The author should give the “AMR data of all bacteria/year” and “distribution of bacteria/year”. The author should do a statistical analysis of this data. I did not find statistical analysis anywhere.

 Abstract:

Please revise and write variables with percentages also (For example 10% (n=10/100). What were the total numbers of samples collected in six years? Among the total, how many samples were positive? Please explain it clearly.

Keywords: First, please write down the name of all bacteria in italic. Secondly, use the full name of the bacteria first i.e., Staphylococcus aureus (S. aureus) and then you can use the short name of the bacteria i.e., S. aureus.

 Introduction:

The introduction is too general: Report the epidemiology of antimicrobial resistance that focus on the study area. Please show the study gaps.

Please write down the introduction in different four paragraphs: 1st: Background 2nd: Study gap, 3rd: Statement of the problem, and 4th: Hypothesis and objective of the study.

 Materials and Methods:

Please rewrite and explain from start to end. How did you collect the samples? What were the conditions applied during sample transportation?  Please make headings of ethical approval and study design, sample collection and transportation, bacterial identification by vitek-2, and so on.

Please give the description of the Hospital. What is the bed capacity? Please refer to the STROBE checklist to ensure each section of the manuscript is well-written (https://www.strobe-statement.org/checklists/).

Provide the manufacturer details of each chemical/material used (Company, City, Country).

 Results:

Not clear. Variables are not cleared and have a lot of difficulties to understand about it.

Section 3.1, Line 82: Please modify this.

What is meant by 1459 microorganisms either positive samples or the total number of samples? Please make it clear.

Table 2: Please give the results into two groups: one for those bacteria isolated after COVID-19 and the second for before COVID and then make a comparison.

Table 2: Gram-positive=675 and gram-negative=655? Please also describe in texts.

Figure 1: Only 7 microorganisms are shown in this figure, but what about the others? Please make a descriptive table in which data on the “distribution of microorganisms of each year” is present.

Section 3.3: Please make an antibiogram of each bacterium.

 Discussion:

The discussion is good. Please revise after making amendments in the result portion.

Revise the discussion accordingly. What did you expect?

Avoid many repetitions of the results, please follow the STROBE guideline.

 Conclusion.

A conclusion is not coming from the results. Please revise this.

Author Response

The submitted manuscript presents “Microbiology and antimicrobial resistance in a tertiary hospital’s surgical ward – a six-year retrospective study”. The author explained the epidemiology of AMR before and after COVID-19, which is good for researchers and physicians to treat infectious diseases and find out alternative therapeutic measures.  The author should give the “AMR data of all bacteria/year” and “distribution of bacteria/year”. The author should do a statistical analysis of this data. I did not find statistical analysis anywhere.

Response: Thanks for the comment. With this revised manuscript we have introduced many changes based on the abovementioned comment of the reviewer. We have made a statistical comparison between the pre-COVID-19 and the post-COVID-19 era showing a statistically significant trend for more Klebsiella and Corynebacterium strains and less Proteus strains in the post-COVID-19 era. Furthermore, we provide detailed data regarding antimicrobial resistance in the supplementary section about the most commonly isolated microorganisms in the present study. We feel that the reader will draw more conclusions while reading the manuscript, and much easier than before.

 Abstract:

Please revise and write variables with percentages also (For example 10% (n=10/100). What were the total numbers of samples collected in six years? Among the total, how many samples were positive? Please explain it clearly.

Response: Thanks for the comment. We have revised the abstract by adding the numbers requested. As also mentioned in the methods section later on, and also added in the abstract, only the positive cultures were included in the analysis of the present study. Thus, the cultures that had not grown any pathogens were unavailable as a number and this information is not provided in the manuscript.

Keywords: First, please write down the name of all bacteria in italic. Secondly, use the full name of the bacteria first i.e., Staphylococcus aureus (S. aureus) and then you can use the short name of the bacteria i.e., S. aureus.

Response: Thanks for the comment. We have turned all names of bacteria in italics (except from Enterobacterales, as correctly noted by another reviewer). We have made some modifications throughout the text so to mention the full gender and species of the microorganisms where applicable (also in the abstract), and then use the first letter of the gender with a dot along with the species when needed. However, in the keywords, we chose to use the gender as a keyword without the species in order to use the broader term available to allow search engines to more easily detect the manuscript for someone who, for example, looks up for Acinetobacter instead of Acinetobacter baumannii.

Introduction:

The introduction is too general: Report the epidemiology of antimicrobial resistance that focus on the study area. Please show the study gaps.

Response: Thanks. We have modified the introduction section according to this and the next comment along with the comments of another reviewer. We now feel that the introduction is more specific and allows the reader to better understand the context of the study.

Please write down the introduction in different four paragraphs: 1st: Background 2nd: Study gap, 3rd: Statement of the problem, and 4th: Hypothesis and objective of the study.

Response: Thanks. We have changed the introduction section according to these comments along with another comment of another reviewer. The introduction is now organized in four paragraphs, with the first giving the background, the second stating the problem, the third providing the study gap and the last one introducing the aim of the present study. This can be seen in the revised version of the manuscript.

Materials and Methods:

Please rewrite and explain from start to end. How did you collect the samples? What were the conditions applied during sample transportation?  Please make headings of ethical approval and study design, sample collection and transportation, bacterial identification by vitek-2, and so on.

Response: Thanks for the comment. We performed the required changes as can now be seen in the revised manuscript in the methods section. Subsections have been added as well as a detailed description of management of the specimens from the time they were taken until pathogen identification and antimicrobial susceptibility testing.

Please give the description of the Hospital. What is the bed capacity? Please refer to the STROBE checklist to ensure each section of the manuscript is well-written (https://www.strobe-statement.org/checklists/).

Response: Thanks for the comment. We looked up to STROBE checklist, and we added some information in the manuscript, and in the methods section in particular, even though this study mostly refers to microbiology, and applicability of STROBE guidelines is partially limited.

Provide the manufacturer details of each chemical/material used (Company, City, Country).

Response: Thanks for the comment. We have added that information regarding all materials used as can be seen in the revised manuscript in the methods section. Details about the manufacturers of culture media and the automated systems used for identification and susceptibility testing have been provided, as recommended by the reviewer.

Results:

Not clear. Variables are not cleared and have a lot of difficulties to understand about it.

Response: We have performed many modifications in the way the results are presented in the revised version of the manuscript to allow the reader to better understand our results. We have changed the Tables, we added dozens of pages in the supplementary material and we now feel that the manuscript is much easier to read and allows the reader to more clearly extract data regarding microbiology and antimicrobial resistance in the surgical ward we studied.

Section 3.1, Line 82: Please modify this.

What is meant by 1459 microorganisms either positive samples or the total number of samples? Please make it clear.

Response: We have modified the methods section to allow the reader to understand that only cultures that were positive were included in the present study. We have now changed that part in the results section as well make it clear at that point as well. This can be seen at that section of the revised manuscript.

Table 2: Please give the results into two groups: one for those bacteria isolated after COVID-19 and the second for before COVID and then make a comparison.

Response: Thanks for the comment. We did the requested analysis and now provide that comparison in the new Table 2 that includes a column with the most common microorganisms in total, a column with the microorganisms isolated in the pre-COVID-19 era, a column showing the microorganisms isolated in the post-COVID-19 era and a last column showing the p resulting from a statistical comparison of the second and the third column. This can be seen in the results section of the revised version of the manuscript. The only statistically significant differences detected had to do with a higher occurrence of Corynebacterium and Klebsiella in the post-COVID-19 era and a lower occurrence of Proteus in the post-COVID-19 era.

Table 2: Gram-positive=675 and gram-negative=655? Please also describe in texts.

Response: We have added that information in the text, as requested by the reviewer, as can be seen in the revised version of the manuscript.

Figure 1: Only 7 microorganisms are shown in this figure, but what about the others? Please make a descriptive table in which data on the “distribution of microorganisms of each year” is present.

Response: These 7 microorganisms consist of 62.1% of strains in total (906 out of 1459 strains), and are also the most common pathogens in the literature as well as in the present study. This is the reason we chose to put them in the figure. We created a table with the information requested, however, we felt it would complicate the manuscript and make it harder for the reader, thus, we put that table in the supplementary material as Table S1. This table summarizes all the data about microbiology in the current study comprehensively. If requested in another revision by the reviewer and the editor, that table could be added in the main body of the manuscript, however, due to reporting of multiple relatively rare pathogens, we felt it would make the manuscript harder for the reader.

Section 3.3: Please make an antibiogram of each bacterium.

Response: Thanks for the comment. Figure 2 was intended to provide the basic information for the most frequently encountered pathogens in the specimens sent from patients hospitalized in a surgical ward. We do agree, however, that providing more information about antimicrobial resistance could be of use for the reader. We doubt that providing information for all the 121 different strains could be of any use for the reader, as most of these strains occurred less than 4 times, thus, providing less than 0.25% of the total number of isolates each. Thus, we decided to provide the requested information in the supplementary table for all microorganisms that were isolated at least 8 times during the six year period of the study, so that they would represent at least 0.5% of the total number of isolates. If the reader needs any more information, we could, upon request, provide the excel file that contains all the information. We don’t feel that providing about 250 pages in the supplementary material could be of use at this point though. The newly added information (that is about 40 pages long) can be seen in the supplementary material of the revised submission.

Discussion:

The discussion is good. Please revise after making amendments in the result portion.

Revise the discussion accordingly. What did you expect?

Avoid many repetitions of the results, please follow the STROBE guideline.

Response: Thanks. We have revised the discussion section and added information regarding the increasing antimicrobial resistance in terms of vancomycin in Enterococcus spp., the relatively stable antimicrobial resistance to other strains, we have added a paragraph regarding microbiology (the most commonly isolated microorganisms) and according to another reviewer’s suggestion, we have added a paragraph about the importance of antimicrobial resistance. We also changed the first paragraph of the discussion section, mentioning the most important findings of the study.

Conclusion.

A conclusion is not coming from the results. Please revise this.

Response:  Thanks for the comment, even though we do not agree completely. The conclusions section in the previous manuscript was a call for a change in the guidelines for empirical treatment in surgical infections. This is not negligible. However, we did change the conclusions section in the revised version of this manuscript by adding more information that came from the newly added results and was also discussed in the discussion section. For example, we comment on the increased isolation VRE strains and the need for action from infection control and antimicrobial stewardship programs. of This can be seen in the revised version of the manuscript.

Round 2

Reviewer 1 Report

1.      The title of the manuscript must be modified special word microbiology.

2.      Replace the word microbiology in the manuscript text with the word microorganisms examples in lines 32, 59, and so on.

3.      Replace the word gender in lines 37 and 158 with the word genus.

4.      Line 35 The most common microbiological sample was pus from surgical wounds rephrase.

5.      Line 44 microbiological data must be correct.

6.      The authors must be added demographic data including age, sex, and so on to patients.

7.      Abbreviations. When used for the first time you write complete and abbreviation between brackets?

8.      Line 123 As quality control strains, Escherichia coli ATCC 25922, Pseudomonas aeruginosa ATCC 27853, S. aureus ATCC 25923, and E. 124 faecalis ATCC 29212. The names of bacterial stains must be written in italic form as the following Escherichia coli ATCC 25922, Pseudomonas aeruginosa ATCC 27853, S. aureus ATCC 25923, and E. faecalis

9.      Line 193 Gram-positive microorganisms correct to Gram-positive bacteria and so on.

10.  Line 142 Gram-negative positive microorganisms correct to Gram-negative bacteria and so on.

Author Response

Reviewer 1

  1. The title of the manuscript must be modified special word microbiology.

Response: Thanks for the comment. We revised the title of the manuscript. The new title is: ‘A six-year retrospective study of microbiological characteristics and antimicrobial resistance in specimens from a tertiary hospital’s surgical ward

  1. Replace the word microbiology in the manuscript text with the word microorganisms examples in lines 32, 59, and so on.

Response: We disagree. The term microbiology is broader. We changed the word microbiology to microbiological characteristics or similar terms where applicable, however, the word microorganisms cannot efficiently replace the word microbiology in any way.

  1. Replace the word gender in lines 37 and 158 with the word genus.

Response: Thanks for the comment. That was a mistake and we corrected that.

  1. Line 35 The most common microbiological sample was pus from surgical wounds rephrase.

Response: Thanks. We changed that to ‘The most common sample sent to the microbiology department was pus from surgical wounds.’

  1. Line 44 microbiological data must be correct.

Response: Line 43-45 contain the following sentence: ‘The antimicrobial resistance rates noted herein raise questions regarding the appropriateness of currently suggested antimicrobials in guidelines, and imply that a revision could be required’. Thus, we do not understand what the reviewer means. We did check, however, the microbiological data above line 43, and they seem to be correct as rates, according to the information in the text (if this was what the reviewer meant).

  1. The authors must be added demographic data including age, sex, and so on to patients.

Response: Thanks for the comment. However, these data are not available. This is why this is a study based on microbiological data. If the requested data were available, we would have already added them. We mentioned that in the first revision as well in a similar comment.

  1. Abbreviations. When used for the first time you write complete and abbreviation between brackets?

Response: Yes. Adding the abbreviation first and the full explanation in brackets is an alternative, however, we feel that there is no right or wrong. Thus, since in all our manuscripts we use this style for abbreviations, we were thinking to keep it that way.

  1. Line 123 As quality control strains, Escherichia coli ATCC 25922, Pseudomonas aeruginosa ATCC 27853, S. aureus ATCC 25923, and E. 124 faecalis ATCC 29212. The names of bacterial stains must be written in italic form as the following Escherichia coliATCC 25922, Pseudomonas aeruginosa ATCC 27853, S. aureus ATCC 25923, and E. faecalis

Response: Thanks. That was a mistake. We corrected that.

  1. Line 193 Gram-positive microorganisms correct to Gram-positive bacteria and so on.

Response: Thanks. That was a mistake. We corrected that throughout the manuscript.

  1. Line 142 Gram-negative positive microorganisms correct to Gram-negative bacteria and so on.

Response: Thanks. That was a mistake. We corrected that throughout the manuscript.

Reviewer 3 Report

Dear Authors,

Thank you for making the necessary changes in paper, please send the paper to English editing to get it easy for readers, you can use editing or any other service for manuscript editing.

Author Response

Thanks for the comments that led to the improvement of the manuscript. The manuscript was read by a native English speaker and corrections were made.

Reviewer 4 Report

The authors have addressed my all suggestions/comments and the revised manuscript looks good.

Author Response

Thanks for the comments that led to the improvement of the manuscript.